# Trans-Atlantic Spillover: Deconstructing the Ecological Adaptation of *Leishmania infantum* in the Americas

**DOI:** 10.3390/genes11010004

**Published:** 2019-12-19

**Authors:** Mariana C. Boité, Gerald F. Späth, Giovanni Bussotti, Renato Porrozzi, Fernanda N. Morgado, Martin Llewellyn, Philipp Schwabl, Elisa Cupolillo

**Affiliations:** 1Laboratory of Research on Leishmaniasis, Oswaldo Cruz Institute, FIOCRUZ, 21040-360 Rio de Janeiro, Brazil; renato.porrozzi@ioc.fiocruz.br (R.P.); morgado@ioc.fiocruz.br (F.N.M.); elisa.cupolillo@ioc.fiocruz.br (E.C.); 2Department of Parasites and Insect Vectors, Laboratory of Molecular Parasitology and Signaling, Institut Pasteur, INSERN U1201, 75015 Paris, France; gerald.spaeth@pasteur.fr (G.F.S.); giovanni.bussotti@pasteur.fr (G.B.); 3Institut Pasteur-Bioinformatics and Biostatistics Hub-C3BI, USR 3756 IP CNRS, 75015 Paris, France; 4Institute of Biodiversity, Animal Health & Comparative Medicine, University of Glasgow, G128QQ Glasgow, UK; martin.llewellyn@glasgow.ac.uk (M.L.); p.schwabl.1@research.gla.ac.uk (P.S.)

**Keywords:** *Leishmania*, visceral leishmaniasis, Americas, genome instability, adaptation

## Abstract

Pathogen fitness landscapes change when transmission cycles establish in non-native environments or spill over into new vectors and hosts. The introduction of *Leishmania infantum* in the Americas into the Neotropics during European colonization represents a unique case study to investigate the mechanisms of ecological adaptation of this important parasite. Defining the evolutionary trajectories that drive *L. infantum* fitness in this new environment are of great public health importance as they will allow unique insight into pathways of host/pathogen co-evolution and their consequences for region-specific changes in disease manifestation. This review summarizes current knowledge on *L. infantum* genetic and phenotypic diversity in the Americas and its possible role in the unique epidemiology of visceral leishmaniasis (VL) in the New World. We highlight the importance of appreciating adaptive molecular mechanisms in *L. infantum* to understand the parasite’s successful establishment on the continent.

## 1. Introduction

*Leishmania infantum*, an etiological agent of visceral leishmaniasis (VL), is a member of the globally dispersed *Leishmania donovani* species complex, prevalent in many regions of the African, American, Asian, and European continents. Despite ongoing debate [1,2,3], it is generally accepted that *L. infantum* was introduced to the American continent during its colonization from Europe beginning ca. five hundred years ago [4,5,6,7,8]. The imported strains were most likely carried by asymptomatic humans, dogs, or other mammalian hosts. Although equally suitable ecological conditions for *L. infantum* transmission are observed in the Old and New Worlds (OW and NW), including the common urban reservoir *Canis familiaris* and the presence of permissive sandfly vector populations, there are important differences that may have uniquely shaped American *L. infantum* genomes (Figure 1). Domestic dogs in European and Latin American countries generally differ in behavior, nutritional status, vaccination, and treatment [9]. Wild canids (e.g., *Lycalopex vetulus* and *Cerdocyon thous)* appear to contribute little to zoonotic transmission in South America whereas the red fox (*Vulpes vulpes*) may represent an important synanthropic host species in Europe. Lagomorphs (comprising hares and rabbits, e.g., *Lepus granatensis* and *Oryctolagus cuniculus*) have also been implicated in suburban outbreaks of disease in Europe [10,11], but infections are seldom reported from South America where marsupials (e.g., the opossums *Didelphis albiventris* in Brazil and *Didelphis marsupialis* in Colombia) may be more relevant to disease transmission from sylvatic to urban environments [12]. Sylvatic host diversity (primates, edentates, rodents, etc.) in the New World is not well described but is likely much higher than in the Old World. With respect to the insect vector of *L. infantum*, the zoonotic transmission cycle in the Mediterranean basin depends on vectors belonging to the *Phlebotomus* genus [13], while the *Lutzomyia* genus represents the vector species for *L. infantum* in Central and South America. These insect genera not only differ in ecological/physiological traits [14] but also, and just as importantly, inter- and intra-species species variances are described [15]. Finally, socio-economic disparities between the New and Old Worlds are relevant for VL, which is associated with poor living conditions and infant malnutrition [12]. 

Visceral leishmaniasis in the Americas now extends from southern parts of the USA [16,17] and Mexico to the North of Uruguay, including countries such as Argentina, Brazil, Paraguay, Bolivia, Venezuela, Suriname, Guyana, Colombia, Honduras, Panama, Costa Rica, El Salvador, Guadeloupe, Guatemala, and Nicaragua [12]. Brazil accounts for 90% of cases [12], and changes in the epidemiological profile have been described over the last 30 years [18]. Initially considered rural, the majority of American Visceral Leishmaniasis (AVL) cases are now observed in peri-urban and urban areas of cities in Northeast, Central-West, and Southeast Brazil [19]. The AVL burden remains very high in the Northeast [20], where the parasites’ rapid adaptation to urban niches has been especially prominent. The first major urban outbreaks occurred in the period 1986–1989 [21] and 1991 [22]. A shift in infection demographics has also become evident. Adult infection rates are increasing while those in children have decreased [23], probably due to improvements in nutrition and immunization rates for other diseases. Atypical cutaneous leishmaniasis caused by *L. infantum* has also been described in the country [24,25]. In such circumstances, the diagnosis based strictly on visual inspection of clinical signs lacking proper *Leishmania* species identification leads to the misassumption of a low risk of VL in the region, further complicating the management of the spread of the parasite and challenging disease surveillance and control. 

VL and dengue are considered the main failures of infectious disease management in Brazil [26]. The observed epidemiological changes and geographic expansion of AVL are associated (but direct causality is yet to be determined) with many variables [27] that compromise the control of this disease [26]. In this review, we describe the underestimated diversity of the neotropical *L. infantum* parasite and its possible role in generating unique epidemiological patterns and clinical outcomes of VL in the New World. We will further comment on the molecular adaptive mechanisms of *Leishmania* and the evolutionary trajectories shaped by region-specific host/parasite genotype-genotype and parasite genotype/environment interactions and discuss their potential impact on the epidemiological scenario in the Americas. 

## 2. Underestimated Genetic Diversity of Neotropical *L. infantum*

The relatively recent arrival of *L. infantum* in the New World is expected to leave a signature of loss of diversity among the neotropical strains. The low mean microsatellite allelic richness of *L. infantum* compared to European *L. infantum* populations indicate such a founder effect [4,8,28]. Lower genetic diversity led to the assumption that any clinical/epidemiological variance observed in the New World should be attributed to host or vector [29] and not to parasite traits. For instance, genetically similar *L. infantum* strains were described, causing dermal (CL) and visceral (VL) leishmaniasis in Honduras [29,30,31], in contrast to descriptions from the Old World. In the Mediterranean region, *L. infantum* parasites also cause cutaneous and visceral disease [32,33,34,35] with significant correlations detected between zymodeme and pathology, as well as genetic differentiation between viscerotropic and dermotropic zymodemes by random amplified polymorphic DNA (RAPD)–PCR fingerprinting [36,37]. However, the low genetic resolution of these approaches compared to those available today likely underestimated the role of parasite diversity in the epidemiology and clinical outcomes of American VL. Moreover, New World sample sets were generally very limited in size (e.g., only four Honduran samples [31]) and spatial extent (representing three Honduran islands [31]), likely further contributing to an assumption of extreme genetic homogeneity among neotropical *L. infantum* populations. In contrast, a more recent study with representative sampling from different Brazilian regions analyzed by random amplified polymorphic DNA (RAPD) showed that, despite the relative genetic homogeneity, most isolates clustered according to geographical origin [28]. Likewise, multilocus microsatellite typing (MLMT) by Ferreira et al. [38] detected substantial population structure in the New World, distinguishing three divergent *L. infantum* populations within Brazil. Such findings call for the application of more resolutive, next generation sequencing (NGS)-based studies to better understand New World *L. infantum* diversity, including genomic structural variation, which is considered a major driver of adaptation in the *Leishmania* genus, as detailed in the following section.

## 3. Mechanisms of *Leishmania* Genomic Adaptation *In Vitro* and *In Vivo*


A hallmark of *Leishmania* biology is represented by the parasite’s capacity to adapt to changing environments despite the largely constitutive expression of its genetic material and the absence of classical transcriptional regulation [39]. The primary mechanisms are little understood even though they may play important roles in driving parasite fitness gain in new reservoirs and host tissues, or in response to drug pressure. Some of *Leishmania*’s adaptative strategies have been described, including the ability to modulate the gene dosage of therapeutic targets (or other determinants of resistance) [40] by chromosomal amplification or the generation of episomal amplicons [41]. A recent series of publications has indeed associated *Leishmania* adaptation to its remarkable genome plasticity, with frequent changes in chromosome and gene copy number generating important intra-strain heterogeneity and complex mosaic populations that are substrates for environmental selection. 

In 2011, Sterkers et al. demonstrated by fluorescence in situ hybridization (FISH) that, in *Leishmania*, mosaic aneuploidy in culture is a common feature [42]. Such karyotypic fluctuations in *Leishmania* have been subsequently confirmed in situ by Barja et al. in amastigotes from infected hamster spleen and liver, where they can drive tissue-specific haplotype selection [43]. These karyotypic variations modulate transcript abundance [39,41,43,44], and, thus, likely generate selectable phenotypes. Longitudinal analyses *in vitro* further demonstrated that the karyoptypic mosaic structure of infecting parasite populations allows for polyclonal adaptation in response to the culture environment, revealing trisomy of chromosome 26 as the major driver of fitness gain in this experimental evolution system [43].

In addition to karyotype variation, read depth analysis of newly established field isolates early after culture adaptation revealed that gene copy number variation (CNV) is a major factor in intra-species genetic heterogeneity and likely drives geographic adaptation. Bussotti et al. [45] evaluated CNVs from the genomes of *L. infantum* field isolates from three continents: South America (Brazil), Europe (Spain), and North Africa (Tunisia). Although evolved in different regions, the genomes were highly homogenous as judged by SNP analysis, suggesting that this kind of genetic variation may have only a little impact on parasite evolution in the field. In contrast, aside from chromosome amplification, the *L. infantum* isolates significant differences in gene CNV, with both gene amplification and gene loss observed for genes previously linked to parasite infectivity, such as GP63 or amastins [45]. These differences could be the result of positive/purifying selection that adapts parasite fitness to a given ecology or transmission cycle [45]. 

Together, these studies reveal highly dynamic fluctuations in chromosome and gene copy number as the major mechanisms allowing for adaptation in the field and in culture. As a result, significant differences exist between tissue-resident amastigotes in patients and derived isolates that are propagated as promastigotes in culture and are subjected to epidemiological association studies. Indeed, significant changes in read depth for 21 chromosomes between *in vitro* passages two and five were demonstrated in fresh field isolates of *L. infantum* from Brazil [45]. The same study showed that strain-specific genomic adaptation is not only governed by the environment, but also by intrinsic genetic factors, as judged by the karyotypic differences observed during *in vitro* adaptation of highly related Spanish *L. infantum* isolates that were cultured under the same conditions. Furthermore, the tissue-origin of parasites may have a similar impact on karyotypic adaptation, as demonstrated by Zackay et al. [46]. The authors reported that strains simultaneously isolated from spleen and skin from the same patient presented different karyotypes and clustered in a tissue-specific way based on aneuploidy profile [46]. These results suggest that genetically different sub-populations of a given isolate are colonizing different organs, with the genetic differences being revealed in culture by distinct adaptive trajectories selected during *in vitro* adaptation.

The available data suggest that chromosome amplification is associated with short-term environmental adaptation to *in vitro* culture, while gain and loss of gene copies might affect long-term environmental adaptation in the field [45]. The detection of alleles under selection in a given region could thus provide relevant epidemiological information. Regarding any field-related inferences, it is important to consider polyclonality with respect to karyotype and gene CNV as an intrinsic feature of any *Leishmania* population. Polyclonal infections are expected to be more difficult for the host’s immune system to control, thus increasing the success of infection. In this way, poly-clonal infections are expected to contribute to parasite fitness gain and virulence in subsequent hosts. [43]. 

Until direct sequencing from tissue is not routinely applied, a useful strategy to infer about the epidemiologically relevant genetic traits of *Leishmania* might be to target CNV signals (amplification, deletion, fixed indels) of fresh field isolates from short-term culture, or purified amastigotes from infected tissues (e.g., from infected dogs). Studying the association of these genotypes with phenotypic traits identified in epidemiological or *in vitro* studies can reveal genetic loci that are under geographic selection. This approach will shed new light on the mechanisms underlying region-specific parasite evolution and clinical manifestations; moreover, it opens new venues for the discovery of biomarkers with diagnostic and prognostic value. 

## 4. Inferences on the Transmission Dynamics of Neotropical *L. infantum* Strains and the Shaping of AVL Epidemiology 

As commented above, until quite recently, *L. infantum* strains from the Neotropics were considered genetically homogeneous, a consequence of either low resolutive typing methods employed and/or poor/biased sampling. Then, in 2012, Ferreira et al. applied microsatellite markers to analyze the population structure of 162 *L. infantum* strains isolated from humans and dogs from most of the Brazilian states endemic for VL and from Paraguay [38]. Samples grouped into three genetically distinct populations: POP1 (relatively more frequent) was observed in all but one endemic area; POP2 was also well-dispersed, but was predominant in Mato Grosso (MT), characterized by the presence of distinct biomes harboring intense biodiversity (Amazon, Cerrado, and Pantanal); POP3 was less dispersed and was observed primarily in Mato Grosso do Sul (MS), likely composed by variable biomes (Cerrado, Atlantic forest, and Pantanal). Several associations with the eco-epidemiological aspects of VL could be identified: First, the distribution of VL in MS seems to follow the west-east construction of the Bolivia-Brazil pipeline from the Corumbá municipality, an inference corroborated by Motoie et al. [47]. Second, regarding vectors, the distribution of POP3 may have been the result of a strong association of this *L. infantum* population and *Lutzomyia cruzi*, which is an important VL vector in the region (Corumbá–MS) [48]. In addition, *Lutzomyia cruzi* also occurs in MT and may influence the structure of POP2 [38,48] as well. The study of Ribolla et al. [49] corroborates that the distinct genetic structure of the sandfly vector plays an important role in shaping the genetic structure of *L. infantum* in Brazil. Quintana et.al. raised the hypothesis of an expanding urban population of *Lutzomyia longipalpis* at the Argentina-Bolivia border, with introgressive hybridisation of older haplogroups found in their path in natural forest or rural environments, acquiring a new adaptability to urban environments, and the possibility of changes in vector capacity [50]. The genetic diversity of *Lu. longipalpis* in Brazil has been recently explored by Casaril et al., who demonstrated the occurrence of three *Lu. longipalpis* populations in endemic municipalities [51]. Such processes of divergence and speciation of vectors represent mechanisms generating the heterogeneity of vector capacity and competence, as well as vector susceptibility to infectious agents or insecticides. 

These studies explored ecological features that possibly shape specific transmission cycles. The question of how the vector/parasite interplay could shape the distribution of *Leishmania* genotypes implies the selection of parasite populations with higher fitness. Such a process involves (among other factors) the variable ability of *Leishmania* strains to attach to the gut and survive the sandfly immune response. The complexities of parasite and sandfly interactions, and their molecular background, have been partially characterized (and revised), but mainly consider only parasites and sandflies species, not genotypes [52,53,54]. For instance, defensin of *Phlebotomus duboscqi* is induced by *L. major* challenge [52], but not by *Leishmania mexicana* in *Lu. longipalpis* [53]. In a more resolutive approach, Mahoney et al. [55,56] assessed intra-species variance, showing polymorphisms in lipophosphoglycan structure controlling *Leishmania donovani*-sandfly interactions. Thus, the question posed for (now recognized heterogeneous) neotropical parasites is: Could *L. infantum* genotype variants represent a fitness gain for survival within the population of vectors from a certain area favoring its spread? Could such successful parasite populations harbor biological differences affecting the interaction with the mammal host, thus also influencing disease phenotypes and transmission patterns? 

These questions were recently addressed by Courtenay et al. [57] in the context of VL in the Indian subcontinent. The authors combined epidemiology with the basic biology of sandflies, parasite, and hosts to infer transmission dynamics. Two modes of sandfly transmission were proposed based on infection dose and its impact on transmission and disease. It was assumed that uninfected insects feeding on asymptomatic vertebrate hosts with low parasitic burden acquire a low number of parasites. This would result in few metacyclic promastigotes to be transmitted during the next blood meal, resulting in a less severe disease. This mode of transmission could explain the maintenance of the parasite in a given population, without severe clinical disease. The alternative possibility would be the feeding of sandflies on a severely infected individual, with a high parasitic load. These insects likely acquire a higher number of parasites during the blood meal and, consequently, many metacyclic promastigotes would be generated. The blood meal on a second host could result in a larger number of metacyclic parasites being transmitted, and this may lead to more acute and severe disease. In both proposed models, the variable of sandfly density also impacts on the dynamics of *Leishmania* transmission [58]. The complexity of this issue was boosted by the recent study of Giraud et al. [59]. The authors have demonstrated that experimental transmission of *L. infantum* by *Lu. longipalpis* is heterogenous (i.e., sandflies can deliver mixed doses of metacyclic and non-metacyclic promastigotes), and this composition is an important determinant of disease outcome and onward transmission. It is plausible, therefore, that genotypic variants of *L. infantum* may differ in their competencies for transmission, which may result in changes in both the dose number and quality, and are worthy of future phenotypic analysis.

The transmission models presented by Courtenay et al., could be templates for projects to explore the specificity of host-parasite-vector relationships and the consequent epidemiological variances in neotropical ecological conditions that shaped the transmission cycles. Genetically and biologically distinct neotropical populations of *L. infantum* could be structured by both the sandfly population distribution and the interplay with mammal hosts, ultimately shaping epidemiological variances of AVL. 

As discussed above, the *Leishmania* “genotype variants” differ not only by the nucleotide sequence-based genotypes (primary sequence) but also, and mainly, in the genomic content, i.e., chromosome and gene copy number variations. To detect this variability, neotropical *L. infantum* genomic diversity has been recently explored by deep sequencing approaches. Teixeira et al. [60] performed a comparative analysis of whole genome sequences of twenty *L. infantum* isolates from humans and dogs in northeastern Brazil. Despite high sequence identity, individual differences displayed sufficient variation to allow the isolates to be clustered based on the primary sequence. A major source of variation detected was in chromosome somy, but the analyses do not suggest that individual sequence variants account for differences in clinical outcome or adaptation to different hosts. The lack of association observed by the authors does not discard the value of somy and gene CNV as informative targets to be coupled with phenotypes. As already mentioned, the plasticity of the *Leishmania* genome could lead to a loss of signal for those samples kept under lab conditions [45]. The karyotypic fluctuations after culture adaptation [45] impacts transcriptomics leading to phenotypic variations [43]. That means that studies that do not use fresh field *Leishmania* isolates, or direct sequencing, might lose relevant signals, unless a strong genomic signal—a result of long-term adaptation, such as gene amplification or deletion [45]—is positively selected. In that circumstance, such signals could eventually be associated with phenotypes or ecological traits. 

This latter possibility is supported by recent work from Carnielli et al. [61]. The authors described a full deletion comprising four genes in tandem on chromosome 31 (the only stable chromosome within the aneuploid mosaicism) of Brazilian *L. infantum* strains. The locus was named ‘miltefosine sensitivity locus’ (MSL) based on the observation that the patients infected with deletion parasites had a 9.4-fold increased risk of treatment failure after miltefosine administration. This deletion further correlated with reduced parasite miltefosine susceptibility *in vitro*, suggesting natural resistance to this drug given that it had never been used in Brazil before this trial was carried out [62]. As a result of the identified correlations, the authors proposed this locus as a potential molecular marker to predict miltefosine treatment outcome in VL. This is of upmost relevance because miltefosine was recently approved in Brazil for treatment of canine visceral leishmanisis (CVL), and little information on the impact of the dog treatment on the epidemiology of AVL is available. Only one study [63], evaluating infected dogs from São Paulo, Brazil, is published. Authors described a remission of clinical signs, reduction in parasitic load by qPCR, and reduction in infectivity to sandflies. However, dogs were evaluated for only a short period, and the *L. infantum* strains were not typed for the MSL locus. If there are indeed non-susceptible miltefosine strains infecting dogs in Brazil, the current treatment policy might represent a risk of worsening the epidemiological scenario of VL in the country, which represents a major public health threat. Nevertheless, a stronger link between MSL and susceptibility to miltefosine is yet to be further explored and confirmed. “How does MSL loss contribute to treatment failure?” is a question already posed in a comment by Bhattacharya and Ouellette [64], which will require much effort to answer. 

The deletion strains described in Carnielli’s work were isolated mainly from patients from northeastern and southeastern Brazil, while all OW strains tested did not present the trait. This pattern indicates fitness gain in response to Brazilian ecological conditions, although more consistent screening of NW and OW strains must be done to better determine frequency and geographic distribution of deleted (Del) and non-deleted (NonDel) samples (Figure 1). The mechanism of loss of the genomic region is possibly mediated by homologous recombination between repetitive elements bordering the locus [61]. Thus, the existence of deleted sub-population among non-deleted *Leishmania* cells cannot be discarded—especially in OW strains, a hypothesis yet to be tested. The possibility of extinction, or reduced frequency of deletion parasites in the OW as a result of changes in environments and socio-economic patterns [65] cannot be disregarded. It is also important to consider the possibility of recombination between the Del and NonDel genotypes, since both were detected co-circulating, and even corresponding to subpopulation within the same isolate [61]. 

Lastly, as already pointed out by other authors [64], the dispersion of deletion parasites indicates that the variant is beneficial, but the nature of the selective advantage driving natural selection of this genomic trait and the selective forces acting within the neotropical ecology remain open questions. The genomic deletion may indeed represent a relevant biomarker, but not necessarily for miltefosine resistance. 

## 5. Concluding Remarks

Parasite expansion into new environments and new host/vector populations creates potential for strong natural selection and bottlenecks, and associated genetic drift can provide the opportunity for rapid evolutionary change. The possibility of major evolutionary change in imported *L. infantum* populations has received little attention. Lack of research in this direction is surprising given the parasite’s extreme medical importance and unique capacity for adaptation based on rapid karyotypic and genetic change. More comprehensive surveys of *L. infantum* genomic and phenotypic diversity across heterogeneous, biodiverse, neotropical environments are required to understand the extensive proliferation of AVL, especially to disentangle demographic from selection events behind complex *L. infantum* population structure and the diverse clinical outcomes observed in the New World. Comparisons between different biomes are especially interesting, e.g., in Central West vs. Northestern Brazil, where different vector species circulate and divergence through natural selection may occur. Alternatively, differences may result from random drift or from distinct parasite importation events. Selection for the distinct imported clones possibly occurred differently, by the diverse environments *Leishmania* parasites were exposed to. Along the years, that possibly led to the founding of specific transmission cycles in the Americas and the respective epidemiological variances observed in the continent. This polyclonal adaptation scenario opens important questions on how the neotropical conditions—with their specific selective pressures (reservoirs, vectors, and other environmental factors) might have shaped New World *L. infantum* evolution. High sampling effort and demographic history reconstruction is therefore also an essential component of future work. Experimental validations of miltefosine resistance represents another priority in the field, as is the potential for putative resistance markers to spread through populations via genetic exchange. To shed new light on the ecology and epidemiology of VL in the Americas, it will be important to combine phenotypic assays with phylogenomic and CNV approaches to trace patterns of demography and selection.

## Figures and Tables

**Figure 1 genes-11-00004-f001:**
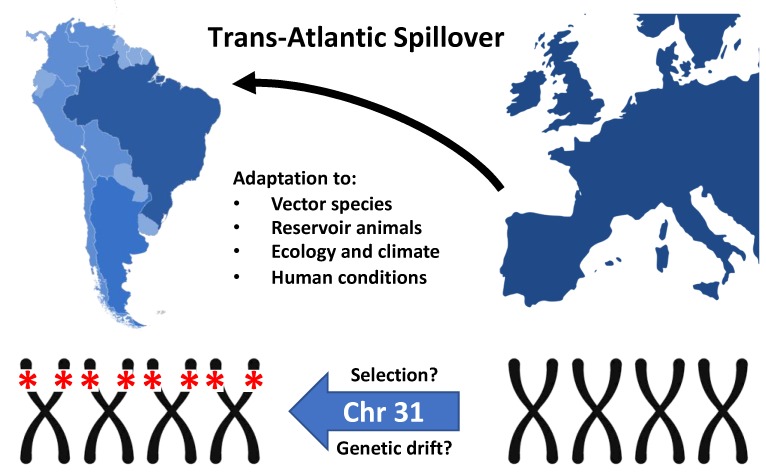
Trans-Atlantic Spillover of *Leishmania infantum* from Southern Europe to the Americas during ‘La Conquista’. The homozygous deletion of four genes from tetrasomic chromosome 31 may either result from genetic drift and expansion of a founder population, or natural selection caused by the encountered, region-specific environmental and ecological conditions.

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
