# Peer review of "Trans-Atlantic Spillover: Deconstructing the Ecological Adaptation of Leishmania infantum in the Americas"

_genes, 2019, doi:10.3390/genes11010004_

Round 1

Reviewer 1 Report

this is an interesting and timely review that can challenge the reader to engage in additional basic research in this topic.  I do recommend that the authors define abbreviations the first time they are used such as line 63 "VL in the Americas (AVL)..."; line 122 gene copy number (GCN) ; lines 273 and 275 OW and NW perhaps define in line 38; line 282:  Del and NonDel not defined

Also, line 71: should dates read 1986-1989?

lines 184 and 186 : define MT and MS

line 217:  Countenay et al. since there is not a single author

line 221:  acquire and not acquires

line 256: use . rather than , in a number, e.g. 9.4-fold

line 291 opportunity misspelled

The format for the Reference Section is not consistent 

Author Response

Thank you very much for your considerations. All comments were revised and the alteartions done.

Best regards, The authors.

Reviewer 2 Report

The authors have put together a really interesting and thought provoking review of the adaptation and recent evolution of L. infantum in the Americas. Although L. infantum is a good example to base such a review, the mechanisms at play are equally applicable to other leishmaniases, so this review should be useful beyond South America.

Major points:

The models of Leishmania infection and transmission by Courtenay et al are interesting ones and seems plausible for L. infantum. Recently, the work of Giraud et al. 2019 (Commun Biol. 2019 Feb 28;2:84. doi: 10.1038/s42003-019-0323-8. eCollection 2019. PMID: 30854476) adds to the complexity of this issue by demonstrating that experimental transmission of L. infantum by Lu. longiplapis is heterogenous (i.e. sand flies can deliver mixed doses of metacyclic and non-metacyclic promastigotes based on the progression of the infection and history of their bloodfeeding) and this can influence both the course of infection and their onward transmissibility. These results are worth highlighting as genotypic variants of L. infantum may have differing competencies for transmission, which may result in changes to both the dose number and quality – and are worthy of future phenotypic analysis.

Throughout the review there is a lot of emphasis on adaptation that has happened and less on what might happen, given that climate change, land use and increasing urbanization is likely to influence AVL in the coming decades. I would like the authors to consider the influence of these on parasite adaptation by bringing modeling studies into their discussion.

A glossary of terminology would be of benefit for those readers not familiar with infectious disease transmission – e.g. sylvatic, synthoropic etc.

Minor corrections:

Line 71: 1986-1989 not 1989-1986

Line 134: Define ‘CNV’

Line 217: Add ‘et al’ to ‘Courtenay’.

Line 256: What is 9,4? Is it 94, 9-4 or 9.4?

Line 275: OW not WO

Line 275: Determine not determining

References: contain e instead of &.

Author Response

Thank you very much for your comments and suggestions. Indeed, we find the mechanisms described applicable to other leishmaniasis. 

We have revised the text adding the minor corrections listed.

We agree with the Reviewer and consider the paper from Giraud et al. 2019 fully relevant for the complexity discussed. A reference to its finding was added. We thank the reviewer for this suggestion.